# Optimal Client Sampling for Federated Learning

## Abstract

It is well understood that client-master communication can be a primary bottleneck in Federated Learning. In this work, we address this issue with a novel client sub-sampling scheme, where we restrict the number of clients allowed to communicate their updates back to the master node. In each communication round, all partici-pated clients compute their updates, but only the ones with "important" updates communicate back to the master. We show that importance can be measured using only the norm of the update and give a formula for optimal client participation. This formula minimizes the distance between the full update, where all clients participate, and our limited update, where the number of participating clients is restricted. In addition, we provide a simple algorithm that approximates the optimal formula for client participation which only requires secure aggregation and thus does not compromise client privacy. We show both theoretically and empirically that our approach leads to superior performance for Distributed SGD (DSGD) and Federated Averaging (FedAvg) compared to the baseline where participating clients are sampled uniformly. Our approach is orthogonal to and compatible with ex-isting methods for reducing communication overhead, such as local methods and communication compression methods.

## 1 Introduction

We consider the standard cross-device Federated Learning (FL) setting [13], where the objective is of the form

$$\min_{x \in \mathbb{R}^d} \left[ f(x) := \sum_{i=1}^{n} w_i f_i(x) \right], \tag{1}$$

where $x \in \mathbb{R}^d$ represents the parameters of a statistical model we aim to find, $n$ is the total number of clients, $f_i \colon \mathbb{R}^d \to \mathbb{R}$ is a continuously differentiable local loss function which depends on the data $\mathcal{D}_i$ owned by client $i$ via $f_i = \mathrm{E}_{\xi \sim \mathcal{D}_i} [f(x, \xi)]$, and $w_i \geq 0$ are client weights such that $\sum_{i=1}^{n} w_i = 1$. We assume the classical FL setup in which a central master (server) orchestrates the training by securely aggregating updates from clients without seeing the raw data.

### 1.1 Communication as the Bottleneck

It is well understood that cost of communication can be the primary bottleneck in Federated Learning. Indeed, wireless links and other end-user internet connections typically operate at lower rates than intra-datacenter or inter-datacenter links and can be potentially expensive and unreliable. Moreover, the capacity of the aggregating master and other FL system considerations impose direct or indirect constrains on the number of clients that are allowed to participate in each communication round. These considerations have led to significant interest in reducing the communication bandwidth of FL systems.

Submitted to 35th Conference on Neural Information Processing Systems (NeurIPS 2021). Do not distribute.

### 1.1.1 Local Methods

One of the most popular strategies is to reduce the frequency of communication and put more emphasis on computation. This is usually achieved by asking the devices to perform multiple local steps before communicating their updates. A prototype method in this category is the Federated Averaging (FedAvg) algorithm [23]. The original work was a heuristic, offering no theoretical guarantees, which motivated the community to try to understand the method and various existing and new variants theoretically [35, 21, 15, 37, 17, 9].

### 1.1.2 Communication Compression

Another popular approach is to reduce the size of the object (typically gradients) communicated from clients to the master. These techniques are usually referred to as gradient/communication *compression*. In this approach, instead of transmitting the full-dimensional gradient/update vector $g \in \mathbb{R}^d$, one transmits a compressed vector $\mathcal{C}(g)$, where $\mathcal{C} : \mathbb{R}^d \to \mathbb{R}^d$ is a (possibly random) operator chosen such that $\mathcal{C}(g)$ can be represented using fewer bits, for instance by using limited bit representation (quantization) or by enforcing sparsity (sparsification). A particularly popular class of quantization operators is based on random dithering [7, 30]; see [1, 41, 42, 28]. A new variant of random dithering developed in [10] offers an exponential improvement on standard dithering. Sparse vectors can be obtained by random sparsification techniques that randomly mask the input vectors and preserve a constant number of coordinates only [40, 18, 36, 24, 39]. There is also a line of work [10, 3] where a combination of sparsification and quantization was proposed to obtain a more aggressive combined effect.

## 1.2 Related Work

Importance sampling methods for optimization have been studied extensively in the last few years in several contexts, including convex optimization and deep learning. LASVM developed in [5], which is an online algorithm that uses importance sampling to train kernelized support vector machines. The first importance sampling for randomized coordinate descent methods was proposed in a seminal paper in [26]. It was showed in [29] that the proposed sampling is optimal. Later, several extensions and improvements followed [33, 20, 6, 27, 2, 38]. Another branch of work studies sample complexity. In [25, 43], the authors make a connection with the variance of the gradient estimates of SGD and show that the optimal sampling distribution is proportional to the per-sample gradient norm. In terms of computation, obtaining this distribution is as hard as the computation of the full gradient, thus it is not practical. For simpler problems, one can sample proportionally to the norms of the inputs, which can be linked to the Lipschitz constants of the per-sample loss function for linear and logistic regression. For instance, it was shown in [11] that static optimal sampling can be constructed even for mini-batches and the probability is proportional to these Lipschitz constants under the assumption that these constants of the per-sample loss function are known. Unfortunately, importance measures such as smoothness of the gradient are often hard to compute/estimate for more complicated models such as those arising in deep learning, where most of the importance sampling schemes are based on heuristics. A manually designed sampling scheme was proposed in [4]. It was inspired by the perceived way that human children learn; in practice, they provide the network with examples of increasing difficulty in an arbitrary manner. In a diametrically opposite approach, it is common for deep embedding learning to sample hard examples because of the plethora of easy non-informative ones [32, 34]. Other approaches use a history of losses for previously seen samples to create the sampling distribution and sample either proportionally to the loss or based on the loss ranking [31, 22]. In [16], the authors propose to sample based on the gradient norm of a small uniformly sampled subset of samples.

In our work, we avoid all the aforementioned problems as our motivation is not to reduce computation, which is not the main bottleneck of Federated Learning, but to *use importance sampling to decrease the number of bits communicated*. This, as we show in Section 2, allows us to construct *optimal adaptive sampling*; that is, we do not need to rely on any heuristics, historical losses, or partial information.

## 1.3 Contributions

In this work, we propose a new approach to addressing the communication bandwidth issues appearing in FL. Our approach is based on the observation that in the situation where partial participation is desired and a budget on the number of participating clients is applied, *careful selection of the participating clients can lead to better communication complexity, and hence faster training.* In other words, we claim that in any given communication round, some clients will have "more informative" updates than others and that the training procedure will benefit from capitalizing on this fact by ignoring some of the worthless updates.

In particular, we propose a principled *optimal client sampling scheme* capable of identifying the most informative clients in any given communication round. Our scheme works by minimizing the variance of the stochastic gradient produced by the partial participation procedure, which then translates to a probable reduction in the number of communication rounds. To the best of our knowledge, this approach was not considered before. Moreover, our proposal is orthogonal to and hence combinable with existing approaches to communication reduction such as communication compression or local updates (Section 3.2).

Our contributions can be summarized as follows:

- we propose a *novel adaptive partial participation strategy for reducing communication in FL* that works by a careful selection of the clients that are allowed to communicate their updates to the master node in any given communication round;

- our *adaptive client sampling procedure is optimal* in the sense that it minimizes the variance of the master update;

- we propose an approximation to our optimal adaptive sampling strategy which only requires aggregation, thus allows for *secure aggregation* and *stateless clients*;

- we show theoretically that our approach allows for *larger learning rates* for Distributed `SGD` and `FedAvg` algorithms than the baseline which performs uniform client sampling, and as a result leads to *better communication complexity.*

- we show empirically that the performance of our approach is superior to uniform sampling and is close to full participation.

## 2 Smart Client Sampling for Reducing Communication

We now describe our client sampling strategy for reducing the communication bottleneck in Federated Learning. Each client $i$ participating in round $k$ computes an update vector $\mathbf{U}_i^k \in \mathbb{R}^d$. For simplicity and ease of exposition, we assume that all clients $i \in [n] := \{1, 2, \ldots, n\}$ are available in each round. However, we would like to point out that this is not a limiting factor, and all presented theory can be easily extended to the case of partial participation with an arbitrary distribution. In our framework, only a subset of clients communicates their updates to the master node in each communication round in order to reduce the number of transmitted bits.

In order to provide analysis in this framework, we consider a general partial participation framework [12], where we assume that the subset of participating clients is determined by an arbitrary random set-valued mapping $\mathbb{S}$ (a "sampling") with values in $2^{[n]}$. A sampling $\mathbb{S}$ is uniquely defined by assigning probabilities to all $2^n$ subsets of $[n]$. With each sampling $\mathbb{S}$ we associate a *probability matrix* $\mathbf{P} \in \mathbb{R}^{n \times n}$ defined by $\mathbf{P}_{ij} := \mathrm{Prob}(\{i, j\} \subseteq \mathbb{S})$. The *probability vector* associated with $\mathbb{S}$ is the vector composed of the diagonal entries of $\mathbf{P}$: $p = (p_1, \ldots, p_n) \in \mathbb{R}^n$, where $p_i := \mathrm{Prob}(i \in \mathbb{S})$. We say that $\mathbb{S}$ is *proper* if $p_i > 0$ for all $i$. It is easy to show that $b := \mathrm{E}\left[|\mathbb{S}|\right] = \mathrm{Trace}\left(\mathbf{P}\right) = \sum_{i=1}^n p_i$, and hence $b$ can be seen as the expected number of clients participating in each communication round. Given parameters $p_1, \ldots, p_n \in [0, 1]$, consider a random set $\mathbb{S} \subseteq [n]$ generated as follows: for each $i \in [n]$, we include $i$ in $\mathbb{S}$ with probability $p_i$. This is called *independent sampling*, since the event $i \in \mathbb{S}$ is independent of $j \in \mathbb{S}$ for any $i \neq j$.

While our client sampling strategy can be adapted to essentially any underlying learning method, we give details here for `DSGD`:

$$x^{k+1} = x^k - \eta^k \mathbf{G}^k, \quad \mathbf{G}^k := \sum_{i \in S^k} \frac{w_i}{p_i^k} \mathbf{U}_i^k, \tag{2}$$

where $S^k \sim \mathbb{S}^k$ and $\mathbf{U}_i^k = g_i^k$ is an unbiased estimator of $\nabla f_i(x^k)$. The scaling factor $\frac{1}{p_i^k}$ is necessary in order to obtain an unbiased estimator of the true update, i.e., $\mathrm{E}_{S^k}\left[\mathbf{G}^k\right] = \sum_{i=1}^n w_i \mathbf{U}_i^k$.

## 2.1 Optimal Client Sampling

We start with a simple observation that the variance of our gradient estimator $\mathbf{G}^k$ can be decomposed as

$$\mathrm{E}\left[\left\|\mathbf{G}^k - \nabla f(x^k)\right\|^2\right] = \mathrm{E}\left[\left\|\mathbf{G}^k - \sum_{i=1}^n w_i \mathbf{U}_i^k\right\|^2\right] + \mathrm{E}\left[\left\|\sum_{i=1}^n w_i \mathbf{U}_i^k - \nabla f(x^k)\right\|^2\right].$$

Note that the second term on the right-hand side is independent of the sampling procedure and the first term is zero if every client sends its update (i.e., if $p_i^k = 1$ for all $i$). In order to provide meaningful results, we restrict the expected number of clients to communicate in each round by bounding $b^k := \sum_{i=1}^n p_i^k$ by some positive integer $m \le n$. This raises the following question: *What is the sampling procedure that minimizes* (3) *for any given* $m$? We answer this question using the following technical lemma:

**Lemma 1.** *Let $\zeta_1, \zeta_2, \ldots, \zeta_n$ be vectors in $\mathbb{R}^d$ and $w_1, w_2, \ldots, w_n$ be non-negative real numbers such that $\sum_{i=1}^n w_i = 1$. Define $\tilde{\zeta} := \sum_{i=1}^n w_i \zeta_i$. Let $S$ be a proper sampling. If $v \in \mathbb{R}^n$ is such that*

$$\mathbf{P} - pp^\top \preceq \mathbf{Diag}(p_1 v_1, p_2 v_2, \ldots, p_n v_n), \tag{3}$$

*then*

$$\mathrm{E}\left[\left\|\sum_{i \in S} \frac{w_i \zeta_i}{p_i} - \tilde{\zeta}\right\|^2\right] \le \sum_{i=1}^n w_i^2 \frac{v_i}{p_i} \left\|\zeta_i\right\|^2, \tag{4}$$

*where the expectation is taken over $S$. Whenever* (3) *holds, it must be the case that $v_i \ge 1 - p_i$.*

It turns out that given probabilities $\{p_i\}$, among all samplings $S$ satisfying $p_i = \mathrm{Prob}(i \in S)$, the independent sampling minimizes the left-hand side of (4). This is due to two nice properties: a) any independent sampling admits optimal choice of $v$, i.e., $v_i = 1 - p_i$ for all $i$, and b) for independent sampling (4) holds as equality. In the context of our method, these properties can be written as

$$\mathrm{E}\left[\left\|\mathbf{G}^k - \sum_{i=1}^n w_i \mathbf{U}_i^k\right\|^2\right] = \mathrm{E}\left[\sum_{i=1}^n w_i^2 \frac{1 - p_i^k}{p_i^k} \left\|\mathbf{U}_i^k\right\|^2\right]. \tag{5}$$

It now only remains to find the parameters $\{p_i^k\}$ defining the optimal independent sampling, i.e., one that minimizes (5) subject to the constraints $0 \le p_i^k \le 1$ and $b^k := \sum_{i=1}^n p_i^k \le m$. It turns out that this problem has the following closed-form solution:

$$p_i^k = \begin{cases} (m + l - n)\frac{\left\|\tilde{U}_i^k\right\|}{\sum_{j=1}^l \left\|\tilde{U}_{(j)}^k\right\|}, & \text{if } i \notin A^k, \\ 1, & \text{if } i \in A^k, \end{cases} \tag{6}$$

where $\tilde{U}_i^k := w_i \mathbf{U}_i^k$, and $\left\|\tilde{U}_{(j)}^k\right\|$ is the $j$-th largest value in $\left\{\left\|\tilde{U}_i^k\right\|\right\}_{i=1}^n$, $l$ is the largest integer for which $0 < m + l - n \le \frac{\sum_{i=1}^l \left\|\tilde{U}_{(i)}^k\right\|}{\left\|\tilde{U}_{(l)}^k\right\|}$ (note that this inequality at least holds for $l = n - m + 1$), and $A^k$ contains indices $i$ such that $\left\|\tilde{U}_i^k\right\| \ge \left\|\tilde{U}_{(l+1)}^k\right\|$. We summarize this procedure in Algorithm 1.

## 2.2 Secure Aggregation

Note that in the case $l = n$, the optimal probabilities $p_i^k = m\frac{\left\|\tilde{U}_i^k\right\|}{\sum_{j=1}^n \left\|\tilde{U}_j^k\right\|}$ can be computed easily: the master aggregates the norm of each update and then sends the sum back to the clients. However, if $l < n$, in order to compute optimal probabilities, the master would need to identify the norm of every

---

**Algorithm 1** Optimal Client Sampling (`OCS`).

---
1: **Input:** expected batch size $m$
2: each client $i$ computes a local update $\mathbf{U}_i^k$ (in parallel)
3: each client $i$ sends the norm of its update $u_i^k = w_i \left\| \mathbf{U}_i^k \right\|$ to the master (in parallel)
4: master computes optimal probabilities $p_i^k$ using equation (6)
5: master broadcasts $p_i^k$ to all clients
6: each client $i$ sends its update $\frac{w_i}{p_i^k} \mathbf{U}_i^k$ to the master with probability $p_i^k$ (in parallel)

---

update and perform partial sorting, which can be computationally expensive and also slightly violates the privacy requirements of clients in FL.

Therefore, we develop an algorithm for approximately solving the problem, which only requires to perform aggregation at the master node without compromising privacy of any client. The construction of this algorithm is similar to [40]. We first set $\tilde{p}_i^k = \frac{m\left\|\tilde{U}_i^k\right\|}{\sum_{j=1}^n \left\|\tilde{U}_j^k\right\|}$ and $p_i^k = \min\{\tilde{p}_i^k, 1\}$. In an ideal situation, this would be sufficient. However, due to the truncation operation, the expected minibatch size $b^k = \sum_{i=1}^n p_i^k \leq \sum_{i=1}^n \frac{m\left\|g_i^k\right\|}{\sum_{j=1}^n \left\|g_j^k\right\|} = m$ can be strictly less than $m$ if $\tilde{p}_i^k > 1$ holds true for at least one $i$. Hence, we employ an iterative procedure to fix this gap by rescaling the probabilities which are smaller than 1, as summarized in Algorithm 2. This algorithm is much easier to implement and computationally more efficient on parallel computing architectures. In addition, it only requires a secure aggregation procedure on the master, which is essential in privacy preserving FL, and thus it is compatible with existing FL software and hardware. We realize that Algorithm 2 brings some extra communication costs, but this is not an issue as it only requires to communicate $\mathcal{O}(j_{\max})$ extra floats for each client. We pick $j_{\max} = \mathcal{O}(1)$, and thus it is negligible for large models of size $d$.

*Remark* 1. We realize that our algorithm requires two communication rounds per optimization round, but the first round is negligible due to the minimal number of communicated bits as argued above.

## 3   Convergence Guarantees

In this section, we provide convergence analysis of `DSGD` and `FedAvg` with our optimal client sampling technique and compare it with full participation and independent uniform sampling of $m$ clients. We use standard assumptions [14] and assume throughout that $f$ has a unique minimizer $x^\star$ with $f^\star = f(x^\star) > -\infty$. We further assume that $f$ is $\mu$-strongly convex and $f_i$'s are $L$-smooth and convex. Detailed definitions of convexity and smoothness can be found in the Appendix. Note that nothing prevents us from extending the results in this section to convex and non-convex cases with a similar standard analysis, since our proposed method only affects the aggregation step as described in Section 2, which is independent of the strong convexity assumption.

**Assumption 1** (Gradient oracle for `DSGD`). The stochastic gradient estimator $g_i^k = \nabla f_i(x^k) + \xi_i^k$ of the local gradient $\nabla f_i(x^k)$, for each round $k$ and all $i = 1, \ldots, n$, satisfies

$$\mathrm{E}\left[\xi_i^k\right] = 0 \tag{7}$$

and

$$\mathrm{E}\left[\left\|\xi_i^k\right\|^2 \mid x_i^k\right] \leq M \left\|\nabla f_i(x^k)\right\|^2 + \sigma^2, \quad \text{for some } M \geq 0. \tag{8}$$

This further implies that $\mathrm{E}\left[\frac{1}{n}\sum_{i=1}^n g_i^k \mid x^k\right] = \nabla f(x^k)$.

**Assumption 2** (Gradient oracle for `FedAvg`). The stochastic gradient estimator $g_i(y_{i,r}^k) = \nabla f_i(y_{i,r}^k) + \xi_{i,r}^k$ of the local gradient $\nabla f_i(y_{i,r}^k)$, for each round $k$, each local step $r = 0, \ldots, R$ and all $i = 1, \ldots, n$, satisfies

$$\mathrm{E}\left[\xi_{i,r}^k\right] = 0 \tag{9}$$

and

$$\mathrm{E}\left[\left\|\xi_{i,r}^k\right\|^2 \mid y_{i,r}^k\right] \leq M \left\|\nabla f_i(y_{i,r}^k)\right\|^2 + \sigma^2, \quad \text{for some } M \geq 0, \tag{10}$$

where $y_{i,0}^k = x^k$ and $y_{i,r}^k = y_{i,r-1}^k - \eta_l g_i(y_{i,r}^k)$, $r = 1, \cdots, R$.

**Algorithm 2** Approximate Optimal Client Sampling (`AOCS`).

1: **Input:** expected batch size $m$, maximum number of iteration $j_{\max}$
2: each client $i$ computes an update $\mathbf{U}_i^k$ (in parallel)
3: each client $i$ sends the norm of its update $u_i^k = w_i \left\| \mathbf{U}_i^k \right\|$ to the master (in parallel)
4: master aggregates $u^k = \sum_{i=1}^n u_i^k$
5: master broadcasts $u^k$ to all clients
6: each client $i$ computes $p_i^k = \min\{\frac{mu_i^k}{u^k}, 1\}$ (in parallel)
7: **for** $j = 1, \cdots, j_{max}$ **do**
8:     each client $i$ sends $t_i^k = (1, p_i^k)$ to the master if $p_i^k < 1$; else sends $t_i^k = (0, 0)$ (in parallel)
9:     master aggregates $(I^k, P^k) = \sum_{i=1}^n t_i^k$
10:     master computes $C^k = \frac{(m - n + I^k)}{P^k}$
11:     master broadcasts $C^k$ to all clients
12:     each client $i$ recalibrates $p_i^k = \min\{C^k p_i^k, 1\}$ if $p_i^k < 1$ (in parallel)
13:     **if** $C^k \leq 1$ **then**
14:         break
15:     **end if**
16: **end for**
17: each clients $i$ sends its update $\frac{w_i}{p_i^k} \mathbf{U}_i^k$ to master with probability $p_i^k$ (in parallel)

We also define two quantities, which appear in our convergence guarantees:

$$R_i := f_i(x^\star) - f_i^\star, \quad r^k := x^k - x^\star, \tag{11}$$

where $f_i^\star$ is the functional value of $f_i$ at its optimum. $R_i$ represents the mismatch between the local and global minimizer, and $r^k$ captures the distance of the current point to the minimizer of $f$.

Equipped with these assumptions, we are ready to proceed with our convergence guarantees. We start with the definition of the improvement factor

$$\alpha^k := \frac{\mathrm{E}\left[\left\| \sum_{i \in S^k} \frac{w_i}{p_i^k} \mathbf{U}_i^k - \sum_{i=1}^n w_i \mathbf{U}_i^k \right\|^2\right]}{\mathrm{E}\left[\left\| \sum_{i \in U^k} \frac{w_i}{p_i^U} \mathbf{U}_i^k - \sum_{i=1}^n w_i \mathbf{U}_i^k \right\|^2\right]}, \tag{12}$$

where $S^k \sim \mathbb{S}^k$ with $p_i^k$ defined in (6) and $U^k \sim \mathbf{U}$ is an independent uniform sampling with $p_i^U = m/n$. By construction, $\alpha^k$ is less than or equal to one, as $\mathbb{S}^k$ minimizes the variance term. In addition, $\alpha^k$ can reach zero in the case where there are at most $m$ non-zero updates. If $\alpha^k = 0$, our method performs as if all updates were communicated. In the worst-case $\alpha^k = 1$, our method performs as if we picked $m$ updates uniformly at random, and one cannot do better due to the structure of the updates $\mathbf{U}_i^k$. In the following subsections, we analyze specific methods for solving the optimization problem (1) under the aforementioned assumptions. The proofs and detailed description are deferred to the Appendix.

**Fairness.** Based on our sampling strategy, it might be tempting to assume that the obtained solution could exhibit fairness issues. In our convergence analysis, we show that this is not the case, as our proposed methods converge to the optimal solution. Hence, as long as the original objective has no inherent issue with fairness, our methods do not exhibit any fairness issues. Besides, our algorithm can be used in conjunction with other "more fair" objectives, e.g., tilted ERM [19].

### 3.1 Distributed `SGD` with Optimal Client Sampling

We begin with the convergence analysis for `DSGD` (see (2)) with optimal client sampling.

---

**Algorithm 3** FedAvg with Optimal Client Sampling.

1: **Input:** initial global model $x^1$, global and local step-sizes $\eta_g^k, \eta_l^k$
2: **for** each round $k = 1, \ldots, K$ **do**
3:     master broadcasts $x^k$ to all clients $i \in [n]$
4:     **for** each client $i \in [n]$ (in parallel) **do**
5:         initialize local model $y_{i,0}^k \leftarrow x^k$
6:         **for** $r = 1, \ldots, R$ **do**
7:             compute mini-batch gradient $g_i(y_{i,r-1}^k)$
8:             update $y_{i,r}^k \leftarrow y_{i,r-1}^k - \eta_l^k g_i(y_{i,r-1}^k)$
9:         **end for**
10:       compute $\mathbf{U}_i^k := \Delta y_i^k = x^k - y_{i,R}^k$
11:       compute $p_i^k$ using Algorithm 1 or 2
12:       send $\frac{w_i}{p_i^k} \Delta y_i^k$ to master with probability $p_i^k$
13:     **end for**
14:     master computes $\Delta x^k = \sum_{i \in S^k} \frac{w_i}{p_i^k} \Delta y_i^k$
15:     master updates global model $x^{k+1} \leftarrow x^k - \eta_g^k \Delta x^k$
16: **end for**

---

**Theorem 2.** *Let $f_i$ be L-smooth and convex for all $i = 1, \ldots, n$. Let $f$ be μ-strongly convex. Suppose that Assumption 1 holds. Choose $\eta^k \in \left(0, \frac{\gamma^k}{(1+\max_{i\in[n]}\{w_i\}M)L}\right)$, where*

$$\gamma^k := \frac{m}{\alpha^k(n-m)+m} \in \left[\frac{m}{n}, 1\right], \quad k = 0, \ldots, K-1.$$

*Define*

$$\beta_1 := \sum_{i=1}^n w_i^2(2L(1+M)R_i + \sigma^2) \quad \text{and} \quad \beta_2 := 2L\sum_{i=1}^n w_i^2 R_i.$$

*Then, the iterates of DSGD with optimal client sampling (6) satisfy*

$$\mathrm{E}\left[\left\|r^{k+1}\right\|^2\right] \leq (1-\mu\eta^k)\mathrm{E}\left[\left\|r^k\right\|^2\right] + (\eta^k)^2\left(\frac{\beta_1}{\gamma^k} - \beta_2\right). \tag{13}$$

**Interpretation.** In order to understand the results of Theorem 2, we first look at the best and worst case scenarios. In the best case scenario, we have $\gamma^k = 1$ for all $k$. This implies that there is no loss of speed comparing to the method with full participation. It is indeed confirmed by our theory as our obtained recursion recovers the best-known rate of DSGD in the full participation regime [8]. Similarly, in the worst case, we have $\gamma^k = m/n$ for all $k$'s, which corresponds to uniform sampling with sample size $m$ and our recursion recovers the best-know rate for DSGD in this regime. This is expected as (12) implies that each update $\mathbf{U}_i^k$ is equivalent, thus we cannot hope for better rate than the uniform sampling. In the general scenario, our obtain recursion sits somewhere between full and uniform partial participation, where the actual position is determined by $\gamma^k$ which capture the distribution of updates (here gradients) on clients. For instance, with a larger number of $\gamma^k$'s tending to 1, we are closer to full participation regime. Similarly, with more $\gamma^k$'s tending to $m/n$, we are closer to the rate of partial participation.

## 3.2 FedAvg **with Optimal Client Sampling**

One of the most common approaches to optimization for Federated Learning is Federated Averaging (FedAvg) [23], an adaption of local-update to parallel SGD. In FedAvg, each client runs some number of SGD steps locally, and then local updates are averaged to form the global update which is then used for the global model on the master. Pseudo-code that adapts the standard FedAvg algorithm to our framework is given in Algorithm 3.

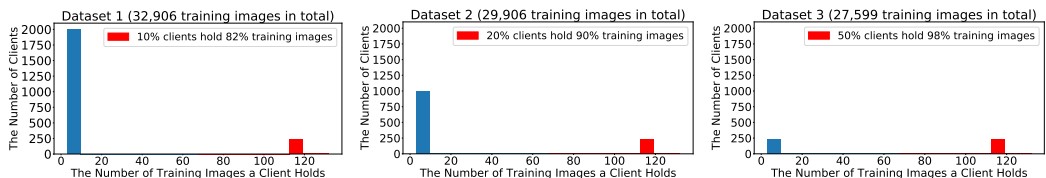

**Figure 1:** Distributions of the three datasets considered.

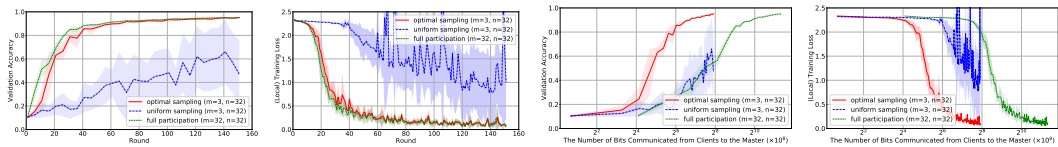

**Figure 2:** (Dataset 1) validation accuracy and (local) training loss as a function of the number of communication rounds and the number of bits communicated from clients to the master.

**Theorem 3.** *Assume that $f_i$ is $L$-smooth and $\mu$-strongly convex for all $i = 1, \ldots, n$ and Assumption 2 holds. Let $\eta^k := R\eta_l^k\eta_g^k$ be the effective step-size and $\eta_g^k \geq \sqrt{\frac{\gamma^k}{\sum_i w_i^2}}$, where*

$$\gamma^k := \frac{m}{\alpha^k(n-m)+m} \in \left[\frac{m}{n}, 1\right].$$

*If $\eta^k \leq \frac{1}{8}\min\left\{\frac{1}{L(2+M/R)}, \frac{\gamma^k}{(1+\max_{i\in[n]}\{w_i\}(1+M/R))L}\right\}$, then the iterates of* `FedAvg` *($R \geq 2$) with optimal client sampling* (6) *satisfy*

$$\frac{3}{8}\mathrm{E}\left[(f(x^k)-f^\star)\right] \leq \frac{1}{\eta^k}\left(1-\frac{\mu\eta^k}{2}\right)\mathrm{E}\left[\left\|r^k\right\|^2\right] - \frac{1}{\eta^k}\mathrm{E}\left[\left\|r^{k+1}\right\|^2\right] + \eta^k\beta_1^k + (\eta^k)^2\beta_2,$$

*where*

$$\beta_1^k := \frac{2\sigma^2}{\gamma^k R}\sum_{i=1}^n w_i^2 + 4L\left(\frac{M}{R}+1-\gamma^k\right)\sum_{i=1}^n w_i^2 R_i \quad and \quad \beta_2 := 72L^2\left(1+\frac{M}{R}\right)\sum_{i=1}^n w_i R_i.$$

**Interpretation.** Similar to `DSGD`, the convergence guarantees of `FedAvg` with optimal client sampling (Algorithm 3) sits somewhere between the performances of those with full and uniform partial participations, where the actual position is again determined by the distribution of updates which directly impact $\alpha^k$'s that are linked to $\gamma^k$'s. In the edge cases, i.e. $\gamma^k = 1$ (best case) or $\gamma^k = m/n$ (worst case), we recover the state-of-the-art complexity guarantees provided in [15] in both regimes. Note that our results are slightly more general, as [15] assumes $M = 0$ and $w_i = 1/n$.

## 4 Experiments

In this section, we empirically evaluate our optimal client sampling method, comparing it with 1) the baseline where participating clients are sampled uniformly from available clients in each round and 2) full participation where all available clients participate. We simulate the cross-device FL setting and train our models using TensorFlow Federated (TFF)[1]. For all three methods, we report validation accuracy and (local) training loss (vertical axis) as a function of the number of communication rounds and the number of bits communicated from clients to the master (horizontal axis). Each figure displays the mean performance with standard error over 5 independent runs. For a fair comparison, we use the same random seed for the three compared methods in a single run and vary random seeds across different runs.

**Setup.** We conclude an evaluation on `FedAvg` where we extend the TFF implementation of `FedAvg`[2] to fit our framework. For the model, we use the two-layer Convolutional Neural Network (CNN)

---

[1] `https://github.com/tensorflow/federated`

[2] `https://github.com/tensorflow/federated/tree/master/tensorflow_federated/python/examples/simple_fedavg`

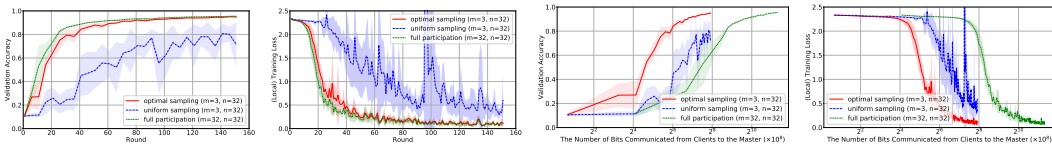

**Figure 3:** (Dataset 2) validation accuracy and (local) training loss as a function of the number of communication rounds and the number of bits communicated from clients to the master.

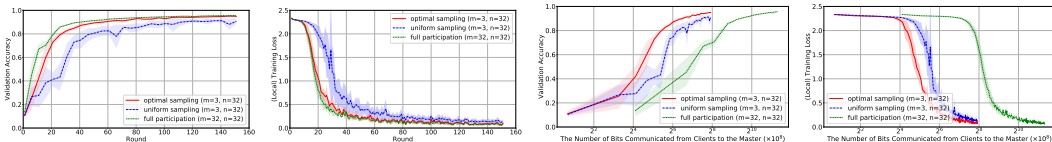

**Figure 4:** (Dataset 3) validation accuracy and (local) training loss as a function of the number of communication rounds and the number of bits communicated from clients to the master.

provided in the implementation. The default dataset is Federated EMNIST with only digits, but as this is a well-balanced dataset with mostly the same quality data on each client, we modify it by removing some clients or some of their training images, in order to better simulate conditions in which our proposed methods bring significant theoretical improvements. As a result, we produce 3 unbalanced datasets as summarized in Figure 1, on which we train the CNN model. For validation, we use the unchanged validation set in the Federated EMNIST dataset, which consists of $40, 832$ validation images. In each communication round of `FedAvg`, $n = 32$ clients are sampled uniformly from the client pool, each of which then performs several SGD steps on its local training images for $1$ epoch with batch size $20$. For partial participation, the expected number of clients allowed to communicate their updates back to the master is set to $m = 3$ for all the experiments. We use constant step sizes, where we set $\eta_g = 1$ and tune $\eta_l$ from the set of values $\{2^{-1}, 2^{-2}, 2^{-3}, 2^{-4}, 2^{-5}\}$ using a holdout set. We implement our sampling procedure using Algorithm 2, as this supports stateless clients and secure aggregation. We include extra communication costs in our results, where we set $j_{\max} = 4$. More details of the hyper-parameters that we use can be found in the Appendix.

**Results and Discussions.** As predicted by our theory, the performance of `FedAvg` with our proposed optimal client sampling strategy is in between the performances of that with full and uniform partial participation. Figures 2, 3 and 4 (red curves: optimal sampling; blue curves: uniform sampling; green curves: full participation) show that, for all three datasets, the optimal sampling strategy performs slightly worse than but is still competitive with the full participation strategy in terms of the number of communication rounds – it almost reached the performance of full participation while only less than $10\%$ of the available clients communicate their updates back to the master. Note that the uniform sampling strategy performs significantly worse, which indicates that a careful choice of sampling probabilities can go a long way towards closing the gap between the performance of naive uniform sampling and full participation.

More importantly, and this was the main motivation of our work, our optimal sampling strategy is significantly better than both the uniform sampling and full participation strategies when we compare validation accuracy as a function of the number of bits communicated from clients to the master. For instance, in case of Dataset 1 (Figure 2), while our optimal sampling approach reached around $85\%$ validation accuracy after $2^6 \times 10^8$ communicated bits, neither the full nor the uniform sampling strategies are able to exceed $40\%$ validation accuracy within the same communication budget. Indeed, to reach the same $85\%$ validation accuracy, full participation approach needs to communicate more than $2^9 \times 10^8$ bits, i.e., $8\times$ more, and uniform sampling approach needs to communicate about the same number of bits as full participation or even more. The results for Datasets 2 and 3 are of a similar qualitative nature, showing that these conclusions are robust across the datasets considered.

In the Appendix, we include additional figures which show the current best validation accuracy as a function of the number of communication rounds and the number of bits communicated from clients to the master.

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
