# OpenReview forum: "Optimal Client Sampling for Federated Learning"
_NeurIPS.cc/2021/Conference — NeurIPS 2021 Submitted_

### Official Review · Reviewer_LDVA · 2021-07-09

**Rating:** 6
**Confidence:** 4

**Summary:**

The paper works on strategies of client subsampling for federated learning. Specifically, it proposes an adaptive partial participation strategy, which can carefully sample clients in each round to sending their updates to the server. More importantly, the paper proposes a practical approximation to optimal sampling strategy. Theoretical analyses are given for the proposed sampling scheme, which shows better performance than existing FedAvg with uniform sampling.

**Limitations And Societal Impact:**

Lemma 1 has been studied in other literature for arbitrary sampling, for example, "S. Horvath and P. Richtarik, “Nonconvex variance reduced optimization ´ with arbitrary sampling,” in International Conference on Machine Learning, 2019, pp. 2781–2789". The authors should cite the related works for reviewers to better position the paper.  What's more, can authors show difficulties and challenges encountered during theoretical analysis?

The theoretical results need the assumption of strong convexity.  However, in experiments, two-layer CNN is used, which is nonconvex. Therefore, in order to be consistent and verify theoretical findings, the authors should either show theoretical results for nonconvex function or run experiments with linear or logistic regression etc.

In experiments, only Fderated EMNIST data is used. The authors should also include more experiments.

**Main Review:**

Originality:  The paper has proposed a client sampling scheme, which shows better convergence performance for federated learning algorithms.  However, the new sampling scheme heavily depends on Lemma 1, which has been well studied in papers for arbitrary sampling.


Quality: The submission technically is sound. Claims are well supported by theoretical analysis or experimental results.

Clarity: The submission is clearly written and well organized.


**Time Spent Reviewing:**

N/A

---

> ### Author Response · Authors · 2021-08-10
> **Response to Reviewer LDVA**
>
> Thank you for your supportive review and valuable time spent reviewing our manuscript.
>
> Concerning the novelty, we do not claim our main contribution to be in the analysis. We claim the **construction of the optimal sampling for Federated Learning client selection (our work is the first to provide such a result) to be the core of our work.** We believe this has substantial merit by itself. In addition, while our analysis might not be entirely novel and might build on the previous results, this does not undermine our contribution, even if the theory might be direct in hindsight for the expert. We do not see that as a reason to remove the merit of our contribution but quite the opposite: the theory in our work aims to describe the properties of the proposed method and not to be a challenge in itself.
>
> With respect to the evaluation, non-convex analysis is possible. It should be straightforward as our main results introduced in Section 2 only affect the aggregation step independent of assumptions such as strong convexity. The simplicity of our approach allows our results to be extended to almost any setting using standard analysis accompanied with our results from Section 2. We will add non-convex analysis in the revised manuscript. Furthermore, we plan to extend our experiments by running on different modality datasets such as the Shakespeare text dataset, although we believe that even in the current form our experiments show strong superior performance while we also argue the same through our theoretical analysis.
>
>
> **We hope that we resolved all the reviewer’s comments and they can reconsider their score.**

---

### Official Review · Reviewer_XEVR · 2021-07-16

**Rating:** 5
**Confidence:** 3

**Summary:**

This paper proposes a strategy to _optimally_ sample clients on a given FL global round. The sampling strategy selects more informative updates by their magnitude. This criterion minimizes the variance of the sampling scheme.

Experiments show this method outperforms uniform sampling.


**Ethical Concerns:**

None.

**Limitations And Societal Impact:**

Yes, this is a theoretical paper and societal impacts,  besides fairness that is addressed in the paper, are out of the scope of this work. I would recommend clarifying and being more specific about the fairness metric used, I am assuming it means just accuracy for all clients? In this case it should be justified why the global optimum is more fair than other solutions.


**Main Review:**


**Originality**
- The paper proposes a nice closed formula deriving a sampling scheme with good properties: minimizing variance. However, it is missing relevant references in the field, particularly the following paper which has a really similar strategy:
--  Rizk, Elsa, Stefan Vlaski, and Ali H. Sayed. "Federated learning under importance sampling." arXiv preprint arXiv:2012.07383 (2020).
This paper seems to derive the same sampling strategy and apply it both to client selection and, within each client, datapoint selection.
- The related work section misses several client sampling strategies in FL.
-- [1] Li, Xiang, et al. "On the convergence of fedavg on non-iid data." arXiv preprint arXiv:1907.02189 (2019).
Chen, Tianyi, et al. "LAG: Lazily aggregated gradient for communication-efficient distributed learning." arXiv preprint arXiv:1805.09965 (2018).
-- Cho, Yae Jee, Jianyu Wang, and Gauri Joshi. "Client selection in federated learning: Convergence analysis and power-of-choice selection strategies." arXiv preprint arXiv:2010.01243 (2020).
-- Ribero, Monica, and Haris Vikalo. "Communication-efficient federated learning via optimal client sampling." arXiv preprint arXiv:2007.15197 (2020).
-- N. Singh, D. Data, J. George, and S. Diggavi. Sparq-sgd: Event-triggered and compressed communication in decentralized stochastic optimization. arXiv preprint arXiv:1910.14280, 2019.

**Quality**
- Line 116 affirms that this method _easily_ extends to an arbitrary availability distribution. This is not vlear to me since previous work has found that, for example under a cyclic distribution, SGD can diverge. How does this extend?
- How to control in practice the number of participant users m?
- How to deal with a large number of clients? In practice not all users can train since the number of clients is in the order order of millions…
- Theorem 2 implies that there is no convergence? since \eta^k is fixed the last term won’t vanish
- Experiments are limited and only compare with uniform sampling and full participation. However, it has been shown in [1] that sampling proportionally to p_i can have better performance. How would this method compare with this sampling strategy?
- The experimental setting is limited and only tested on variations of EMNIST. It would be interesting to see how this sampling behaves in different tasks.

**Clarity**
- The introduction is clear but as it advances the paper could improve the phrasing, it is easy to get lost in the convergence guarantees.
- Small comments:
-- In line 170 is it p_is that are larger or smaller?
-- Line 227 It should be “our obtained”. In general this paragraph needs some editing, there are several typos.

**Significance**
-  In certain FL settings, client sampling can improve the accuracy of the model while reducing communication. This paper proposes a sampling scheme theoretically motivated. However, the proposed strategy (or similar ones mentioned above) have already been proposed by previous work. I would recommend the authors to clearly state how their work differentiates from these works that select clients for training - particularly approaches relying on update norm and/or importance sampling.


**Time Spent Reviewing:**

3

---

> ### Author Response · Authors · 2021-08-10
> **Response to Reviewer XEVR**
>
> We would like to thank the reviewer for the feedback and valuable time spent reviewing our manuscript.
>
> **Concerning the originality of the paper:**
>
> 1. We would like to kindly ask the reviewer to drop the first comment about the originality of our work with respect to the work of  Rizk et al. Please note this is a recent unpublished paper and appeared several months after our arXiv submission, which we can’t point to due to double-blind reviewing policy. The referenced paper should be criticised if it produces the same result later by several months.
>
> 2. Regarding Li et al., this work provides analysis of FedAvg under uniform client sampling and, therefore, it is only tangentially relevant to the question of the optimal sampling. Regarding the works for event-triggered communication (Chen et al., Ribero & Vikalo and Singh et al.), each client communicates only if their update is larger than some threshold, independent of other clients updates. We believe that these are orthogonal approaches that one could combine, e.g., only clients that satisfy communication triggers communicate their norm to the master. Regarding the most relevant work of Yae Jee et al., this is an unpublished concurrent work that appeared on arXiv simultaneously as ours. Furthermore, they require control over client sampling at the beginning of each communication round, and the Power-Of-Choice algorithm can’t be executed using only Secure Aggregation. In summary, we are happy to add these works into the related work section, but we also note that this is a minor issue.
>
> **Regarding quality:**
>
> 1. Thank you for raising this. We mean that as long as the given algorithm converges for a given availability distribution, we can extend the convergence results to employ our optimal sampling strategy on top of this algorithm. It is unrealistic to assume that our sampling procedure could fix non-convergent availability distributions such as the one mentioned by the reviewer.
>
> 2. $m$ depends on the capacity of the master, i.e., how many clients updates it can handle during the aggregation step without incurring high latency.
>
> 3. We note that full participation is restrictive due to practical considerations. To approximate clients availability in the real-world setting in our experiments, we first sample $n$ clients from a pool of clients in each round.  Then, we select  $m$ important clients from these $n$ clients to communicate updates to the master. Please note that we can easily extend the full availability setting to partial participation, as mentioned in the manuscript (lines 114-117).
>
> 4. We note that this term also appears in standard SGD analysis and is due to the stochasticity of gradients, and it cannot be removed. To obtain convergence to arbitrary precision, one can gradually decrease the step size.
>
> 5. Thank you for suggesting this baseline. This might be indeed a stronger baseline than the uniform sampling. On the other hand, we note that such a method does not improve when the global function’s weights are not proportional to the number of data points and can be worse than simple uniform sampling. We also note that if each client runs a local epoch instead of local steps with the same step size, the norm of the update likely correlates with the number of local data points. We will consider adding the proposed baseline in the future version of our work.
>
> 6. We plan to extend our experiments by running on different modality datasets such as the Shakespeare text dataset, although we believe that even in the current form our experiments show strongly superior performance, while we also argue the same through our theoretical analysis.
>
> Regarding clarity, we thank the reviewer for the suggestion to improve our presentation. We plan to address all the comments in the revised version.
>
> **To conclude, we believe we addressed all your concerns, especially the ones concerning the novelty of our work. Therefore, we would kindly request the reviewer to reconsider their score.**

---

> > ### Comment · Reviewer_XEVR · 2021-08-26
> > **Thanks for the detailed response.**
> >
> > I think the paper is really interesting and I like how the algorithm is derived from a nice closed formula.
> >
> > However, I believe clarity about related work, and how the paper builds on top of this or compares to other related work is missing. I believe the authors claim is true that the work I mention is concurrent, but I still would suggest including a comment on this, pointing out it is a concurrent (posterior?) work. But there is still other references that are not clearly discussed. I agree that theoretical papers contributions is more in the analysis and not experiments, but given that it is not clear how this work builds on [43], a clear discussion on the novelty is still required.
> >
> > Further, the experimental section, where some FL particularities could differentiate the potential of this work in FL settings, is limited compared to typical FL papers that test at least on two or more datasets and tasks.
> >
> > For these reasons I am maintaining my score.

---

> > > ### Author Response · Authors · 2021-08-29
> > > **Thank you for the extra feedback.**
> > >
> > > We want to thank the reviewer for their response to our rebuttal, and we hope we can further address the reviewer's concerns.
> > >
> > > Regarding [43], we want to clarify that [43] is only concerned and argues about optimality in the case $m=1$. Our work extends their optimality result to any $m = 1, 2, \dots, n$, which is a non-trivial extension and is not a simple consequence of [43]. We realise that we don't specifically mention this difference, and we will add this discussion to the revised manuscript. We hope that this resolves the issue about a missing proper comparison to [43]. Regarding other references, we note that this is a minor issue that we can easily address by mentioning these works as concurrent posterior work as proposed by the reviewer.
> > >
> > > We are also concluding extra experiments that we hope to be finished by the camera-ready deadline.

---

### Official Review · Reviewer_DC2U · 2021-07-16

**Rating:** 6
**Confidence:** 4

**Summary:**

The authors propose an efficient client (sub) sampling scheme for federated learning (FL) which relies on the quality of the update of each (sampled) client. The authors justify their construction by designing the (sub) sampling strategy that minimizes the gradient variance over the whole network (or sampled clients). The authors analytically show the convergence of DSGD and FedAvg under their sampling strategy which improves on the uniform sampling strategy. Finally, the authors empirically validate the proposed sampling algorithm by showing considerable communication gains compared to both and uniform client sampling strategies.

**Limitations And Societal Impact:**


yes

**Main Review:**


Communication is a bottleneck for large-scale FL applications and the design of efficient sampling strategies is, therefore, an important problem. The authors consider the sampling design problem from the viewpoint of minimizing the variance of the estimated gradient. The paper is well written and the presentation is sufficiently clear. The experiments presented in the paper show that the proposed sampling algorithm can reduce the communication requirements of the standard algorithms considerably compared to the baselines. Below some of the concerns and comments are listed which the authors should address, especially, the technical concerns:

1.	The discussion from lines 148 to 149 is not clear. Specifically, it is not clear why the authors claim the optimality of the proposed client sampling strategies. In inequality (4), the authors present an upper bound of the client sampling strategy and show that i) the upper bound is minimized for the independent sampling strategy (via the choice of $v_i$’s) and ii) the inequality will be equality for independent sampling. It seems to me that the above-discussed sampling strategy will minimize the upper bound on the r.h.s. of equation (4) rather than minimizing the variance of the sampled gradient. Please make the above discussion clear and verify the claims.

2.	The first equation in Section 2.1 will hold if the expectation of the inner product between the two terms $G^k - \sum_{i=1}^n w_i U_i^k$ and $ \sum_{i=1}^n w_i U_i^k - \nabla f(x^k)$ is zero. This will happen only if conditioned on  $U^k_i$ the estimate $G^k$ is unbiased (the probabilities $p_i^k$ are dependent on $U^k_i$ as from Algorithms 1 and 2 the distribution of $S^k$ changes with $p_i^k$). Please clarify this in the paper. It would be helpful if the authors can explain the expectations.

3.  The goal of the paper is to minimize the communication requirement but the authors only try to minimize the gradient variance as the authors also state in Lines 93-95 that their scheme “probably” leads to communication reduction. So, I am not sure if calling the proposed sampling strategies optimal is justified.

4.	The authors should give intuition behind the approximate Algorithm 2, specifically, why it will approximately minimize the variance. Moreover, it appears that Algorithm 2 can be fully executed at the server which might alleviate some of the communication requirements.

5.	For FedAvg (Algorithm 3), the $U_i^k$ used for sampling the clients is different from the one discussed in Section 2. Do the proposed analysis and the discussion of Section 2 still work, and again, can the authors claim optimality for their sampling strategy?

6.	FL algorithms as well as the experiments authors conduct are on non-convex losses. So, it would be interesting to see the performance of the proposed sampling strategies for non-convex losses.

7.	The approach proposed by the authors is closely related to event-triggered distributed algorithms. The authors should discuss the advantages/drawbacks compared to event-triggered problems.

Minor Comments:
1.	In Theorem 2 strong convexity of $f_i$’s is not required, however, in Theorem 3 it is required. Why?
2.	Discussion in Line 203: “at most $m$ non-zero updates” is not clear
3.	In Line 166-167, what is the ideal situation authors are referring to?
4.	In the 4th contribution bullet the authors say that a larger learning rate leads to better communication complexity. Why is that so? Because of faster convergence? Also, if relevant please discuss this in detail with the main results.
5.	Please correct, Line 142: “the sampling procedure that minimizes (3)”
6.	In the abstract line 4: correct participated to participating
7.	In the abstract, instead of “formula” probably use “expression”
8.	Line 22: $\mathcal{D}_i$ is data or distribution of data. Also, define $\xi$.
9.	Please add some references to the claims made in Section 1.1 of the Introduction.
10.	Section 1.2: Lines 58-59 are not clear. It was shown in [29] that the sampling in [26] is optimal? Please clarify
11.	Line 83: replace “addressing” by “address”
12.	Update the discussion from Lines 148-151
13. Is it standard to have the dependence of the step-sizes on $\alpha^k$?

-----------
Updated the score.

**Time Spent Reviewing:**

8

---

> ### Author Response · Authors · 2021-08-10
> **Response to Reviewer DC2U**
>
> Thank you for your valuable feedback, suggestions and time spent reviewing our manuscript.
>
> **Regarding your main concerns:**
>
> 1. Your concern about optimality is indeed correct, and we acknowledge that the reviewer raises a fair point. As the reviewer mentioned, we need independence or sample only a single client for the inequality to hold equality to guarantee optimality with respect to the actual variance rather than its lower bound. Please note that requiring independent sampling accommodates the privacy requirements of Federated Learning. Furthermore, we can show that (4) is tight, and therefore, if one is allowed to communicate norms (1 float per client) as side information, then our sampling is optimal in a general sense. Of course, we cannot guarantee optimality in a fully general setting. The actual optimal sampling that minimises variance can’t be obtained without revealing, i.e. communicating, all the updates to the master. We will clarify this in the revised manuscript.
>
> 2. Please note that this is precisely the setting in which we operate. While we define a general partial participation distribution in lines 120-130, we realise that it is essential that we require this distribution to be unbiased. Thank you for the comment. We will clarify this.
>
> 3. Our algorithm achieves communication reduction by importance sampling that minimises the gradient variance, which directly minimises the convergence rate as we show in our convergence theorems and therefore minimises communication due to a decrease in the number of steps the algorithm needs to converge. We use the word “probable” because no method could do better than uniform sampling in extreme cases where all updates are the same.
>
> 4. Intuition is greedily refining the raw probabilities to obtain probabilities defined in (6), as discussed in lines 164-175. As correctly noted, indeed, Alg. 2 can be fully executed on the server. However, in this case every client would need to reveal their update norm (line 12), which can’t be performed using Secure Aggregation. In addition, if the update norm was revealed, the master could use Alg. 1. On the other hand, we wouldn’t need the extra communication rounds if there was a way to execute line 12 privately on the server.
>
> 5. Yes, even though $U_i^k$ can’t be linked to gradient as in Section 2, we still obtain optimal sampling regarding minimising the variance of partial participation and optimising convergence.
>
> 6. The proposed extension is possible since our main results introduced in Sec. 2 only affect the aggregation step independent of assumptions such as strong convexity. The simplicity of our approach allows our results to be extended to almost any setting using standard analysis accompanied with our results from Sec. 2. We will add non-convex analysis in the revised manuscript.
>
> 7. In event-triggered communication, each client communicates only if their update is larger than some threshold, independent of other clients' updates. We believe that these are orthogonal approaches that one could combine, e.g., only clients that satisfy communication triggers communicate their norm to the master. We are happy to add event-triggered communication into the related work section.
>
> **Minor comments:**
>
> 1. It is standard to require this stronger assumption for FedAvg as it is generally more complicated to analyse.
>
> 2. We mean that at least $n - m$ of the $U^i_k$ vectors are zero.
>
> 3. Ideal situation: Formula in line 166 is equivalent to (6).
>
> 4. Yes, as in our case, larger step sizes lead to faster convergence. We will rewrite this and make it explicit that we mean faster convergence.
>
> 5.-12. Thank you, we will fix these.
>
> 13. Yes, this is standard as our method can be seen as an adaptive method since our sampling is not fixed but depends on the current updates from clients.
>
> **We hope that we addressed all your comments sufficiently, and if you agree, we would kindly request an increase in score in light of our response. If you do not agree, please let us know why, and we will be happy to respond further.**

---

> > ### Comment · Reviewer_DC2U · 2021-08-23
> > **Further comments**
> >
> > I thank the authors for addressing the comments.
> >
> > Overall, the paper is well written but the authors should improve/clarify the following:
> >
> > - The introduction section should be updated by adding other relevant references as pointed out by reviewers.
> > - The authors must clearly discuss the optimality issues in the revised version because the claim of optimality (by minimizing the upper bound on the gradient variance) makes the title and the paper a little misleading.
> > - Since the authors claim that analysis for non-convex losses is straightforward, it would be better if they add the analysis and discussions for non-convex losses.
> >
> > I am willing to increase the score if the above points are addressed.
> >
> > In my opinion, the strongest point of the paper is the experiment section where the authors show massive gains compared to uniform sampling. However, one question I have (which other reviewers also raised) is are there other baselines that the authors can compare their approach to?

---

> > > ### Author Response · Authors · 2021-08-24
> > > **Thank you for your comments, we will clarify the mentioned points**
> > >
> > > Thank you for your comments. We will incorporate them into the revised manuscript.
> > >
> > > Regarding your first bullet point, we will update the references as mentioned by the reviewers.
> > >
> > > Regarding your second point, we will clarify the optimality issue as discussed in our first response. We note that if one assumes sampling to be independent across the clients, our upper bound holds equality, and therefore our optimality results hold as claimed. We further note that assuming independence is aligned with standard Federated Learning privacy requirements. We will add a discussion that clarifies the aforementioned points.
> > >
> > > Regarding non-convex analysis, we provide analysis of Distributed SGD in the non-convex setting in the following comment.
> > >
> > > We would kindly request from the reviewer whether this comment with the non-convex analysis of Distributed SGD addresses their points or more discussion is needed.

---

> > > > ### Author Response · Authors · 2021-08-24
> > > > **Non-convex analysis of Distributed SGD with Optimal Client Sampling**
> > > >
> > > > Throughout this analysis, we use the notation introduced in our manuscript. In several places, we also refer to the equations from the manuscript.
> > > >
> > > > For the analysis of Distributed SGD with Optimal Client Sampling in the non-convex setting, we assume that Assumptions 1 and $f $ is a Lipschitz smooth with smoothness constant $L$. We further assume that similarity among local gradients.
> > > >
> > > > $$
> > > >   \sum_{i=1}^n w_i \|\|  \nabla f_i(x) -  \nabla f(x) \|\|^2 \leq \rho
> > > > $$
> > > >
> > > > We note that this is a standard assumption for the analysis of Distributed SGD in the non-convex setting.
> > > >
> > > > Using (2), the fact that $\mathbf{G}^k$ is an unbiased estimator of $\nabla f$ and Lipschitz smoothness of $f$, we obtain that.
> > > >
> > > > $$
> > > > E \left[ f(x^{k+1})\right] \leq E \left[ f(x^{k+1})\right]  - E \left[ \eta^k \|\| \nabla f(x^k) \|\|^2 \right]  + E \left[ (\eta^k)^2 \|\| \mathbf{G}^k \|\|^2 \right].
> > > > $$
> > > >
> > > > To upper bound $E \left[ \|\| \mathbf{G}^k \|\|^2 \right]$ , we reuse equalities provided in lines 534 and 535, i.e.,
> > > >
> > > > $$
> > > > E \left[ \|\| \mathbf{G}^k \|\|^2 \right] \leq E\left[ \left((1 + M)\alpha^k \frac{n-m}{m} + M \right) \sum_{i=1}^n w_i^2 \|\| \nabla f_i(x^k)\|\|^2 \right] + E\left[ \left(\alpha^k \frac{n-m}{m} + 1 \right) \sum_{i=1}^n w_i^2 \sigma^2 \right]  + E \left[ \|\| \nabla f(x^k) \|\|^2 \right].
> > > > $$
> > > >
> > > > We further bound $ \sum_{i=1}^n w_i^2 \|\| \nabla f_i(x^k)\|\|^2$ using the similarity assumption
> > > >
> > > > $$
> > > >  \sum_{i=1}^n w_i^2 \|\| \nabla f_i(x^k)\|\|^2 \leq \max_i w_i \sum_{i=1}^n w_i \|\| \nabla f_i(x^k)\|\|^2 =  \max_i w_i \sum_{i=1}^n w_i \|\| \nabla f_i(x^k) - \nabla f(x^k) \|\|^2 + \|\| \nabla f(x^k) \|\|^2 \leq \max_i w_i \rho +   \max_i w_i \|\| \nabla f(x^k) \|\|^2
> > > > $$
> > > >
> > > > Combining the aforementioned equations leads to
> > > > $$
> > > > E \left[ f(x^{k+1})\right] \leq E \left[ f(x^{k+1})\right]  - E \left[ \eta^k\left( 1 - \frac{L}{2}\eta^k \beta^k \right) \|\| \nabla f(x^k) \|\|^2 \right]  + E \left[(\eta^k)^2 \gamma^k\right],
> > > > $$
> > > > where $\beta^k = (1 + M)\left(\alpha^k \frac{n-m}{m} + 1 \right) $ and $ \gamma^k = \left((1 + M)\alpha^k \frac{n-m}{m} + M \right) \max_i w_i \rho + \left(\alpha^k \frac{n-m}{m} + 1 \right) \sum_{i=1}^n w_i^2 \sigma^2$. The above inequality is the standard form of one step recursion for obtaining convergence results in the non-convex regime.
> > > >
> > > > We hope that this result resolves the reviewer concern about the applicability of our proposed sampling in the non-convex regime.

---

### Official Review · Reviewer_xb8E · 2021-07-19

**Rating:** 5
**Confidence:** 3

**Summary:**

The paper proposes to use importance sampling in client sampling in federated learning to improve convergence behavior and reduce communication.
Contribution:
1. A new importance sampling scheme based on the norm of gradients/updates.
2. Experiments show the proposed sampling scheme outperforms uniform sampling.

**Limitations And Societal Impact:**

Yes.

**Main Review:**

Strengths:
1. Theoretical optimality of the new client sampling method is given to motivate it.
2. The proposed client sampling method outperforms uniform client sampling.

Weaknesses:
1. The major concern is there is no comparison with other importance sampling methods. There is a large amount of literature on importance sampling methods for optimization algorithms to improve convergence, the proposed importance sampling scheme should be properly compared with existing literature and highlight the novelty of the proposed scheme. For example, [1] is a well-known importance sampling scheme for SGD to reduce update variance, the sampling probabilities depend on the magnitude of gradient (which will become update magnitude if trivially adapted to federated learning setting). The authors have a one-sentence argument that the difficulty of using [1] is the complexity is equivalent to using the full-batch gradient, which I think is because estimating the sampling probabilities requires computing all per-sample gradients. However, in the authors' setting, estimating sampling probabilities require estimating updates from all clients or at least some selected clients, though it might be practical, why can't one apply the importance sampling method in [1] with the sampling candidates being all clients or the selected clients (instead of the per-sample gradient)? Also, I feel the importance sampling technique in [1] is quite similar to the proposed importance sampling method, though the proposed scheme has some special adaptation to distributed settings (Algorithm 2). In a word, my concern is that the novelty of the proposed sampling scheme is unclear compared with other existing importance sampling methods for SGD. I hope the authors could add a more clear discussion on this point especially the difference and improvement compared with [1] since the sampling scheme is the main contribution.

2. There is not much in-depth analysis on the improvement of the proposed client sampling method over uniform client sampling. The convergence analysis relates the convergence rate with \alpha^k (12) but the discussion on how \alpha^k changes with gradient/update distribution are very limited.  I think the most important thing is to get an in-depth understanding of how the reduction of variance depends on gradient/update distribution, which is not given enough attention in the paper. Instead, the paper focus on the convergence analysis of algorithms. However, in my impression, many convergence analysis of SGD and FedAvg directly depends on the variance of the update (I am pretty sure this is the case for SGD, but for some advanced sharp analysis for FedAvg it might be different), with importance sampling applied, when the estimator is unbiased, the only thing that is changed in the convergence analysis could be the variance term, so the new analysis could be just directly substituting the variance term into the existing analysis which is kind of trivial. If the convergence analysis is non-trivial, it could be great to point out the difficulties in analysis when using importance sampling to highlight the challenges and avoid giving the readers an impression of trivialness.

Summarizing my two points, I think the novelty of the proposed scheme is not clear to me. I have an impression that the proposed importance sampling scheme is not much different from traditional importance sampling methods for SGD and the convergence analyses are not much different from existing ones. If my impression is true, it makes the contribution of this paper being the application of importance sampling in FedAvg which I believe is not enough to pass the acceptance bar for NeurIPS. I am willing to change my score if the authors could convince me of the novelty of the paper.

[1]. Zhao, Peilin, and Tong Zhang. "Stochastic optimization with importance sampling for regularized loss minimization." international conference on machine learning. PMLR, 2015.

**Time Spent Reviewing:**

3

---

> ### Author Response · Authors · 2021-08-10
> **Response to Reviewer xb8E**
>
> We thank the reviewer for their feedback and valuable time spent reviewing our manuscript!
>
> Concerning the main criticism, we would like to stress that what the reviewer claims to be a weakness is actually a vital feature of our paper. In the related work section, we already exactly addressed the reviewer’s comment, discussing in-depth the standard importance sampling strategies. The main concern is, as the reviewer points out, that we do not compare against other methods, especially the one proposed in Zhao & Zhang (2015) ([43] in our paper). We would like to point out that this is a misunderstanding. Our optimal sampling strategy is *identical* to the optimal sampling proposed in [43]. We acknowledge this by thoroughly reviewing approaches that mimic this optimal sampling strategy by different approximations as in their setup of interest; this procedure is inefficient, e.g. sampling based on smoothness constants in [43]. The identification that *FL setup allows to employ the optimal sampling without any approximation because communication is a bottleneck* is the main novelty of our work (Lines 79 - 83 and Section 2), and there is no previous work that could achieve this. In contrast, the plethora of works focusing on communication reduction for FL is large.
>
> To your second point, we believe this is well addressed in lines 199-208 and below each convergence theorem, where we discuss how $\alpha^k$ affects the convergence. The analysis is as challenging as obtaining SGD/FedAvg with arbitrary client participation regarding the convergence analysis. We would like to stress that there is no such analysis for FedAvg, making our contribution non-trivial. Furthermore, although the theory might be direct for the expert (SGD case), we do not see that as a reason to remove the merit of our contribution but quite the opposite - the theory in our work aims to describe the properties of the proposed method and not to be a challenge in itself.
>
> We hope that we sufficiently addressed your comments. If this is not the case, please can you clarify what was left unaddressed? We would be happy to respond.

---

> > ### Comment · Reviewer_xb8E · 2021-08-24
> > **Thank you for the reply but my concerns are not addressed**
> >
> > I would like to thank the authors for providing a point to point reply to my review. However, my concerns are not addressed.
> >
> > 1. Since the author acknowledged that their method is identical to the optimal sampling proposed in [43], this should be clearly stated in the paper to avoid giving readers an impression that the optimal sampling is proposed by the authors. The true contribution of the paper is applying the existing optimal sampling method in FL setting with modifications. The whole page 3 is devoted to discuss why the sampling is optimal without mentioning [43]. Since the proposed approach is exactly from [43], similar optimality theories should be already provided in [43] and should be cited.
> >
> > 2. Thanks for referring to line 199-208 but I would like the authors know that I already read these lines before I wrote my comment. In my opinion, these lines are just a very vague discussion that is not sufficient. I want to make my point clear that since the contribution is the client sampling, at least some theorems should be given focusing on the sampling variance reduction instead of the convergence theory. The authors acknowledged that some convergence theories are straightforward for expert and do not want to remove them, this is totally fine. My point is some theory on sampling variance should be provided to give more insights instead of just providing ones that might be straightforward for experts.

---

> > > ### Author Response · Authors · 2021-08-24
> > > **Our results are not identical to [43]**
> > >
> > > We apologise for not being precise in our arguments before.
> > >
> > > Regarding [43] and our sampling, we want to clarify that [43] is only concerned and argues about optimality in the case $m=1$. In our work, we extend their optimality result to any $m  = 1, 2, \dots, n$, which is a non-trivial extension and is not a simple consequence of [43], therefore this criticism is invalid. We realise that we don't specifically mention this difference, and we will add a section discussing this difference. We hope that this resolves the issue about a missing proper comparison to [43].
> > >
> > > Addressing the second concern, we realise that lines 199-208 might be just a vague discussion about the actual improvement. This is because the sampling variance depends on the actual updates which are unknown and, in general, do not follow the specific distribution, therefore proper analysis is impossible. We would be grateful if the reviewer has any suggestions about what kind of theory they would like us to provide. One idea that we can imagine is to provide a theory under the assumption of the update distributions. We are happy to add such results if the reviewer believes that this might give more insights to readers.

---

### Decision · Program_Chairs · 2021-09-27

**Decision:**

Reject

**Comment:**

In this work, the authors propose an efficient client sampling scheme for federated learning (FL) which relies on the quality of the update of each (sampled) client. The sampling strategy is designed based on  minimizing the gradient variance over clients. The authors provided convergence analysis for the DSGD and FedAvg algorithms under the proposed sampling strategy. Finally, the authors showed that numerically, considerable communication gains can be achieved compared  the uniform client sampling strategies.

Pros:

1) A new importance sampling scheme has been introduced to the FL setting, which is based on the norm of gradients/updates. Theoretical optimality of the new client sampling method is given to motivate it.

2) Convergence analysis has been conducted.

3) Experiments show the proposed sampling scheme outperforms uniform sampling.


Cons:

1) Many reviewers have pointed out, that the authors have not made a fair comparison with [43]. The optimal sampling scheme is strongly motivated by the sampling scheme in [43], but the authors fail to point this out in the paper. This has caused a lot of confusion about the main contribution of the paper.

2) Analytical improvement of the proposed client sampling method over uniform client sampling is not clear.


3) Claims about extensions, for example to client intermittency, non-convex losses, have been mentioned, but these claims are somewhat misleading since they should be either analytically or empirically justified.

4) Finally, despite the fact that large gains have been shown, the empirical result is a bit limited compared to typical FL papers (which include multiple datasets and tasks).

Overall, the most important reason for rejecting this paper is that, the AC and the reviewers remain not convinced about the main contribution of this work. I would suggest the authors to take these comments into consideration, and thoroughly compare their scheme with [43], and clearly demonstrate that their approach is non-trivial extension of the existing optimal sampling scheme.